# COVID-19: A Comparative Study of Contagions Peaks in Cities from Europe and the Americas

**DOI:** 10.3390/ijerph192416953

**Published:** 2022-12-16

**Authors:** Karine Bertin, Johanna Garzón, Jaime San Martín, Soledad Torres

**Affiliations:** 1Centro de Investigación y Modelamiento de Fenómenos Aleatorios-Valparaíso, Instituto de Ingeniería Matemática, Facultad de Ingeniería, Universidad de Valparaíso, Valparaíso 2362905, Chile; 2Departamento de Matemáticas, Facultad de Ciencias, Universidad Nacional de Colombia, Bogotá 111321, Colombia; 3Centro de Modelamiento Matemático, Departamento de Ingeniería Matemática, Unité Mixte Internationale, Centre National de la Recherche Scientifique, Universidad de Chile, Santiago 8370456, Chile

**Keywords:** COVID, cluster analysis, regression model

## Abstract

Coronavirus disease 2019 (COVID-19) is an infectious disease caused by a group of viruses that provoke illnesses ranging from the common cold to more serious illnesses such as pneumonia. COVID-19 started in China and spread rapidly from a single city to an entire country in just 30 days and to the rest of the world in no more than 3 months. Several studies have tried to model the behavior of COVID-19 in diverse regions, based on differential equations of the SIR and stochastic SIR type, and their extensions. In this article, a statistical analysis of daily confirmed COVID-19 cases reported in eleven different cities in Europe and America is conducted. Log-linear models are proposed to model the rise or drop in the number of positive cases reported daily. A classification analysis of the estimated slopes is performed, allowing a comparison of the eleven cities at different epidemic peaks. By rescaling the curves, similar behaviors among rises and drops in different cities are found, independent of socioeconomic conditions, type of quarantine measures taken, whether more or less restrictive. The log-linear model appears to be suitable for modeling the incidence of COVID-19 both in rises and drops.

## 1. Introduction

Coronavirus disease 2019 (COVID-19) is an infectious disease caused by a group of viruses called coronaviruses that provoke illnesses ranging from the common cold to more serious illnesses such as pneumonia, Middle East respiratory syndrome (MERS) and severe acute respiratory syndrome (SARS). The coronavirus strain (COVID-19) that has caused the outbreak in China is novel and was not previously known; it has been rapidly exported to other countries and has severely affected different aspects of human health worldwide. On 11 march 2020, the World Health Organization declared the COVID-19 outbreak an epidemic. The COVID-19 virus spread rapidly from a single city to an entire country in just 30 days and to the rest of the world in no more than 3 months. The rapid geographic spread and the sudden increase in the number of cases surprised and quickly overwhelmed the health and public health services of several countries. Hence, governments in different countries promoted border closures, social distancing, school closures and other mandatory control measures in order to reduce the spread of the virus.

Several studies have tried to model the behavior of COVID-19 in diverse regions. Different models based on differential equations of the SIR and stochastic SIR type (and their extensions) have been applied (ref. [1]) and statistical comparisons of the COVID-19 dynamic, especially that of the governmental measures behavior, from different countries have been done (see [2,3,4,5]).

Classical susceptible–infectious–removed (SIR) epidemic models have been widely used in the literature to model infectious diseases. These mathematical models are structured according to the number of susceptible (S), infectious (I) and removed (recovered or deceased) individuals (R). The dynamic behavior of these models depends mainly on the basic reproduction number. The disease is controlled and subsequently eliminated if the basic reproduction number is less than one and continues to spread otherwise [6]. However, in practice, infectious diseases may exhibit periodic oscillations and other nonlinear phenomena during the outbreak. Therefore, several studies have attempted to model this nonlinear behavior.

In this article, another direction was taken, by comparing the speed to reach different contagion peaks and, respectively, the decline in different cities. These were chosen because they provided easy access to their data and because of the type of containment measures adopted by their governments. These cities were Bogota, Lima, Lyon, London, Madrid, Mexico City, Marseille, Paris, New York, Rome and Chile Metropolitan Region.

Several articles compare the impact of COVID-19 infections and mortality in different countries. For example, Ref. [7] studied the early impact from February to July 2020 in the Nordic nations of Sweden, Norway, Denmark and Finland using publicly available case data sources. Ref. [8] studied countries from Asia and Europe in terms of cases, death and case fatality rate. Ref. [9] investigated the spreading rate of COVID-19 comparing 10 European countries. In relation to the comparison of cities by COVID-19, Ref. [10] examined the impacts and outcomes of COVID-19 in three cities, New York, London and Tokyo, taking COVID-19-related deaths in these cities and their respective countries as a variable. Some differences were found, possibly explained in part by political, socioeconomic and cultural differences.

In this paper, we wanted to compare 11 cities to each other, in terms of their downward/upward slopes at different contagion peaks. Since these cities had different population sizes, the number of new daily cases were not on the same scale. Moreover, these data were highly nonlinear. Therefore, the data were normalized by applying a natural logarithm function and a log-linear regression model was applied.

## 2. Material and Methods

This study used different data sources. In Section 2.1, information about the chosen cities and the data studied is presented, while Section 2.2 explains how the data were modeled.

### 2.1. Dataset

A dataset was created containing daily infected cases of COVID-19 between 1 February 2020 and 30 August 2021 for eleven cities. Note that the data from all the cities were not exactly from the same dates. Table 1 shows the eleven cities and the abbreviations used, as well as the respective repositories.

In the sequel, for each city, i=1,…,11, $N_{i}\left(t\right)$ denotes the daily number of infected cases of COVID-19 detected in city *i* at day *t*, where the series of infected cases detected have been smoothed with a moving average of 7 days to eliminate weekend variations. Figure 1 shows the daily number of infected cases for these cities up to 24 April 2021, which corresponds to the last day in which all cities had information.

As it can be observed in Figure 1 and Figures in Appendix A, depending on the city, the number of infected had one, two or three peaks. Table 2 summarizes the number of observed rises and drops for each city. In fact, most of the cities had at least 2 peaks (2 upward and 2 downward movements) whereas London (LO) and Mexico City (ME) had only 1 peak.

To compare the development of different cities, with diverse control measures, it was necessary to normalize them to the maximum observed at each peak. More specifically, for i=1,…,11 and j=1,2,3, $A_{ij}$ was defined as the maximal value of $N_{i}$ for city *i* and peak *j* and the day $T_{ij}$ where this maximum was reached. In fact, note that $N_{i}\left(T_{ij}\right)=A_{ij}$. For each city *i*, peak *j* and day *t*, denote by ${\left(CN\right)}_{ij}\left(t\right)=N_{i}\left(t-T_{ij}\right)/A_{ij}$, the centered and normalized data. Figure 2 shows the normalized and centered data for each city and the corresponding peaks. In this way, each curve of cases in different cities and peaks were overlapped on the same graph, so that the peaks coincided at time 0 (time translation). Thus, −50 (days) represented, for each curve, 50 days prior to the corresponding peak. The curves were also normalized in height, so that the peak of each curve was 1. That is, each curve of cases was divided by its own maximum (normalization of cases). In this way, different curves at different locations, which had different sizes, could be compared to see if they responded to a similar mechanism. It can be seen that the normalized curves had similar behaviors even when distinct containment measures were taken.

In Section 2.2, two regression models for the logarithm of these normalized data, centered around their peaks, are proposed. In these models, increasing and decreasing slopes are estimated. Our goal is then to compare the cities in terms of these estimated slopes (see cluster analysis in Section 2.3 and Section 2.4).

### 2.2. Regression Models

In this section, two regression models are proposed. The first one models the total number of rises and drops. The second one models the rises and drops in a neighborhood of the peak, that is, between the peak and the days when a city exceeds half of the corresponding peak.

#### 2.2.1. The *Total* Regression Model

A previous step to the definition of the *total* model is the definition of the beginning of the rises and the end of the drops. Our model for the rises considered the starting point of a rise as the point at which a significant change in the slope was observed with respect to the pseudo-steady state. The same reasoning was applied for the end of a drop. For example, for the city of Bogota, Figure 3 plots the cases between February 2020 and August 2021. Increases are identified with circles and drops with asterisks. Sometimes, an asterisk is superimposed on a circle. Three rises and drops can be isolated. Figure 4 and Figure 5 show the rises and drops of the logarithm of daily reported cases. For the eleven cities, the daily reported cases and the rises and drops can be found in Appendix A.

Denote by *R* a rise and by *D* a drop, for the *total* model (full rise and drop). For a city *i* and a peak *j*, $T_{ij}^{R}$ denotes the starting point of a rise and $T_{ij}^{D}$, the end point of a drop. Recall that $T_{ij}$ has been defined in Section 2.1 and it is the point where the maximum value of the peak is reached. The following model is then considered
(1)$$\begin{array}{c} \log\left(\frac{N_{i}\left(T_{ij}^{R}+s\right)}{N_{i}\left(T_{ij}\right)}\right)=\alpha_{ij}^{R}+\beta_{ij}^{R}s+\epsilon_{i,j}^{R}\left(s\right),\;s=0,\cdots ,T_{ij}-T_{ij}^{R}; \end{array}$$
(2)$$\begin{array}{c} \log\left(\frac{N_{i}\left(T_{ij}+s\right)}{N_{i}\left(T_{ij}\right)}\right)=\alpha_{ij}^{D}+\beta_{ij}^{D}s+\epsilon_{i,j}^{D}\left(s\right),\;s=0,\cdots ,T_{ij}^{D}-T_{ij}, \end{array}$$
where $\epsilon_{i,j}^{R}\left(s\right)$ and $\epsilon_{i,j}^{D}\left(s\right)$ are independent white noises. Models (Equation 1) and (Equation 2) are log-linear regression models. Estimates for the slope parameters $\beta_{ij}^{R}$, $\beta_{ij}^{D}$ and the intercept parameters $\alpha_{ij}^{R}$ and $\alpha_{ij}^{D}$ were obtained using the classical least square estimators. The estimators of the upward and downward slopes are given in Table 3 together with their goodness of fit.

Note that the starting and end points of the rises and the drops were not entirely well-defined, as can be seen in the pictures of the Appendix A. Therefore, the model could vary significantly due to this starting point. This was one of the reason for considering the *50%* model defined in Section 2.2.2, which did not require obtaining these points.

#### 2.2.2. The *50%* Regression Model

Denote by *r* a rise and by *d* a drop for the *50%* model. Moreover, $T_{ij}^{r}=\max\left\{t<T_{ij}:\;N_{i}\left(t\right)\leq 0.5N_{i}\left(T_{ij}\right)\right\}$ is defined as the day that the city *i* exceeds half of the corresponding peak *j*. In a similar way, $T_{ij}^{d}=\min\left\{t>T_{ij}:\;N_{i}\left(t\right)\leq 0.5N_{i}\left(T_{ij}\right)\right\}$ is defined as the first day after the peak *j* that $N_{i}$ falls below half of the peak.

Our goal was to study the relationship between time and the series $N_{i}\left(T_{ij}^{r}+s\right):s=0,\cdots ,$

$T_{ij}-T_{ij}^{r}$ and $N_{i}\left(T_{ij}+s\right):s=0,\cdots ,T_{ij}^{d}-T_{ij}$. The following model was considered
(3)$$\begin{array}{c} \log\left(\frac{N_{i}\left(T_{ij}^{r}+s\right)}{N_{i}\left(T_{ij}\right)}\right)=\alpha_{ij}^{r}+\beta_{ij}^{r}s+\epsilon_{i,j}^{r}\left(s\right),\;s=0,\cdots ,T_{ij}-T_{ij}^{r}; \end{array}$$
(4)$$\begin{array}{c} \log\left(\frac{N_{i}\left(T_{ij}+s\right)}{N_{i}\left(T_{ij}\right)}\right)=\alpha_{ij}^{d}+\beta_{ij}^{d}s+\epsilon_{i,j}^{d}\left(s\right),\;s=0,\cdots ,T_{ij}^{d}-T_{ij}, \end{array}$$
where $\epsilon_{i,j}^{r}$ and $\epsilon_{i,j}^{d}$ are independent white noise processes. In the same way as done previously, least square estimators were calculated for the intercepts and the slopes (see Table 4). For the rises, the highest slope was observed for the first rise of Madrid at 0.11 along $|T_{ij}-T_{ij}^{r}|=6$ days. The lowest slope was for the second rise of Marseille at 0.00066 during $|T_{ij}-T_{ij}^{r}|=82$ days, but with an $R^{2}$ close to zero. Most of the models had a good $R^{2}$, several of them close to or greater than 0.9. Only 4 had values of $R^{2}$ lower than 0.7. Those below this value were the second rise of RM, the third rise of MD, the second rise of MR and the first rise of ME. With respect to the drops, the slopes took values between −0.068 (first drop of LY) and −0.0068 (second drop of NY). Only one case had an $R^{2}<0.6$, the second drop of MR. The limiting case corresponded to the first drop of Lima with $R^{2}=0.56$.

There were some drops, in which the number of reported cases failed to fall below 50% and subsequently restarting a new rise, a phenomena that we called the *wadding* effect. In the figures given in Appendix A, this can be seen in the following cases: 3 BO goes up (Figure A2); 1 LI goes down (Figure A6); 1 and 2 MR go up (Figure A16); 1 ME goes up (Figure A19); 2 NY goes down (Figure A22); 2 RM goes up (Figure A27); 1 RO goes down (Figure A31).

In the second Marseille descent, two types of descents were observed: initially a slower one and then a steep one. Moreover, the descents were in general more regular than the rises.

### 2.3. Upward Slopes in the *Total* and *50%* Models

In this section, the slopes are studied in both regression models. Cities were classified according to the upward slopes estimated in the linear regression models. Table 5 gives the estimates for the upward slopes and the number of days $|T_{ij}-T_{ij}^{q}|$ with q∈{r,R}.

From Table 5, it can be seen that in general, the second ascents in the *50%* model were slower than or at least equal to the first one with the exception of MD and PA. The cases of RM and NY were distinguished because in the second peak there was a smaller slope in a greater number of days, which could be interpreted as a lower number of infected per day during the second peak than during the first one. Additionally, the slopes in the *total* model were in general larger than the slopes in the *50%* model. The 50th percentile was considered to be a central point where quarantine or other confinement measures began to result in a slope change. Finally, one way to check that the model had a good fit was whether the product Days × Slope was close to log(2)≈0.69. Notice this quantity should be close to log(2), because it represents the logarithmic increase from the 50% to the peak.

In Figure 6, the slopes of the rises in the *total* and *50%* models are plotted. It is clear that the first rises of Madrid (MD) in the *total* model (0.277) and at *50%* (0.115) were atypical (outliers) points in the graph. So the cluster analysis of the upward slopes was performed without the first rises of Madrid.

A cluster Analysis (CA) is one of the data mining techniques that groups the sample observations into classes depending on the essential similarities within a class and the dissimilarities between the different classes found in the dataset. A cluster analysis has been used in the context of the COVID-19 epidemic for comparing countries in terms of mobility and cases (see [11]). In this work, a cluster analysis based on a K-means method and a squared Euclidean distance was performed using Matlab software.

Figure 7 suggests that three was the optimum number of clusters for each model (*total* and *50%*), and Table 6 shows the three clusters and the corresponding means.

The classes were obtained by classifying the different rises by the means from lowest to highest, indicating the speed of growth of the number of infected. A difference was evident when considering the *total* model and the *50%* model slopes. The means of the classes with the totality of the data were higher than the means of the classes with the 50% data, which could be attributed to a saturation phenomenon. Several rises from Cluster 1 with the full data moved into Clusters 2 or 3 with respect to the 50th percentile data. NY remained in Cluster 3 with the highest data growth rate. With the exception of LO, the Cluster 2 rises with the entire data remained in Cluster 2 of the 50th percentile data.

The 50th percentile model’s rises had three behaviors or classes, namely: slow (Cluster 1), medium (Cluster 2), and fast (Cluster 3). However, in the total model’s rises, there were essentially two behaviors and an outlier rise in NY, probably due to a faulty measurement, which was also evident in the differences of the slopes (see Figure 7).

### 2.4. Downward Slopes in the *Total* and *50%* Models

In this section, downward slopes are studied in both regression models. Cities were classified according to the downward slopes observed in the log-linear regression models. Table 7 contains the results for the downward slopes.

Observe that the first slopes were larger in absolute value than during the second peak. RM exhibited a similar behavior in the first and second downslope. The same occurred in RM with the number of days. In BO and LI, an increase occurred in the slope of the second peak and the number of days decreased. In the rest of the cities, the slope decreased, and the number of days increased. This increase in the second dip could be interpreted as the measures taken in the second phase not having much effect or being weaker or less respected. The product −Days × Slope in several runs was close to log(2), which indicated that the model fitted the data well.

Figure 8 also suggests that three was the optimum number of clusters for both models (*total* and *50%*), and Table 8 shows the three clusters and the corresponding means.

As with the rises, drops were classified (by the K-means method) into three groups with both *total* and *50%* models. Several drops moved from being in Cluster 2 with the total data to Cluster 1 with 50% of the data. Only MD and LI moved from Cluster 1 with the total data to Cluster 2 with 50% of the data.

## 3. Discussion

In this article, a simple statistical analysis of the new coronavirus disease 2019 (COVID-19) outbreak in 11 cities from Europe and America was provided. Using daily infected data in the cities for approximately the months after the first cases were confirmed in each city, downward and rising slopes for each wave were analyzed using a log-linear model approach. These models were consistent with the rises and drops of SIR-type models if one thinks about each of the pandemic waves.

This study was triggered by the question of whether southern hemisphere cities had anticipated measures learned from the experience of northern hemisphere cities that had their first peaks earlier.

We found that the rises and drops were almost independent of the type of quarantine measures taken, whether more or less restrictive. By rescaling the curves, similar behaviors between the rises and drops of the different cities were obtained. For example, there were similarities between cities such as NY and RM, independently of their socioeconomic conditions. The question that remains here is whether this has self-similarity within each city. Of course, the height of the peaks strongly depended on several reasons, namely climatic conditions, confinement measures, testing intensity among many others. However, the evidence provided by the data showed that when the curves of detected cases were rescaled, they had an unexpected regularity and similarity between different cities of the world with disparate socioeconomic conditions, with very dissimilar health systems and where very distinct containment measures were applied.

In this direction, Figure 9 and Figure 10 shows the social distancing and the use of mask in the nine countries considered (http://www.healthdata.org/covid/data-downloads, accessed on 3 February 2022, data from the corresponding cities were used if available; see the explanation of how the data were collected on the web page); in both figures the horizontal axis is the time measured in days (one corresponds to 2 April 2020). Figure 9 shows the evolution of the social distancing, measured through cell phone data, in different geographical areas, showing a higher variability (except for the first peak) than our normalized curves. The zero on the vertical scale represents the social distancing before the pandemic. A negative number represents the reduction in the social distancing. For example, an index of −80% means a reduction in 80%. It is interesting to note how this distancing evolved upwards after the first wave, with smaller declines in the periods of the successive peaks. This disparity was also observed in the use of face masks (see Figure 10) in public places for the different geographical areas. The vertical axis represents the percentage of mask use in different countries. To this must be added the high unpredictability of the evolution of COVID-19, as shown in Canals et al. [12], where a comparison of the main Lyapunov exponent was made on the same data series for different countries, obtaining that these varied between 5 and 15, much higher than in other epidemics such as N1H1, where the main coefficient was of the order of 2.

When applying the log-linear regression model to the daily infected cases, the results for the increases in the 11 chosen cities revealed three groups differentiated by the corresponding slopes. For the *50%* model, a higher growth rate for group three and a lower one for group one, between 0.001 and 0.007 for the first, 0.017 and 0.026 for the second and 0.029 and 0.036 for the third. When considering the total data, the results showed three groups also differentiated by a higher growth rate for group three and a lower one for group one, between 0.004 and 0.036 for the first one, 0.045 and 0.061 for the second one and the third one containing a single point or outlier with a slope of 0.122.

For the downslope phases of the outbreaks, the results in the 11 chosen cities revealed three groups differentiated by a higher growth rate for group three and lower for group one, between −0.007 and −0.016 for the third, −0.03 and −0.041 for the second and −0.058 and −0.068 for the *50%* model. When considering the total data, the results showed three groups also differentiated by a higher growth rate for group three and a lower one for group one, between −0.008 and −0.035 for the third, −0.042 and −0.059 for the second, and the first one containing a single point or outlier with a slope of −0.078 which corresponded to 1 LY.

## 4. Conclusions

The log-linear regression models studied for the rises and drops showed that different cities had a similar behavior, independent of their socioeconomic situations. On the other hand, for the same city, very different behaviors were observed from one peak to another.

These results show that estimates can only give reasonable indications in the short term and that other variables such as health interventions, the level of distancing and other public policies are necessary in any model that wants to predict in the medium term.

Despite the simplicity of these models, they provide an interesting insight into the COVID-19 statistics in 11 cities in Europe and the Americas, allowing comparisons to be made between them. The log-linear model appears to be suitable for modeling the incidence of COVID-19 both in rises and drops. In addition, the results could be useful for supporting health policy decisions or government interventions. However, they should be used in conjunction with other more complex mathematical and epidemiological models.

## Figures and Tables

**Figure 1 ijerph-19-16953-f001:**
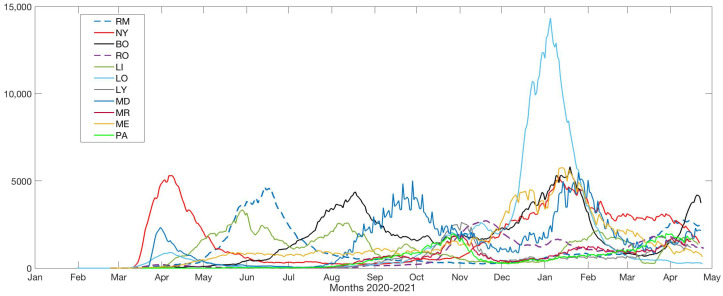
Daily changes of coronavirus disease 2019 cases in different cities during 2020–2021.

**Figure 2 ijerph-19-16953-f002:**
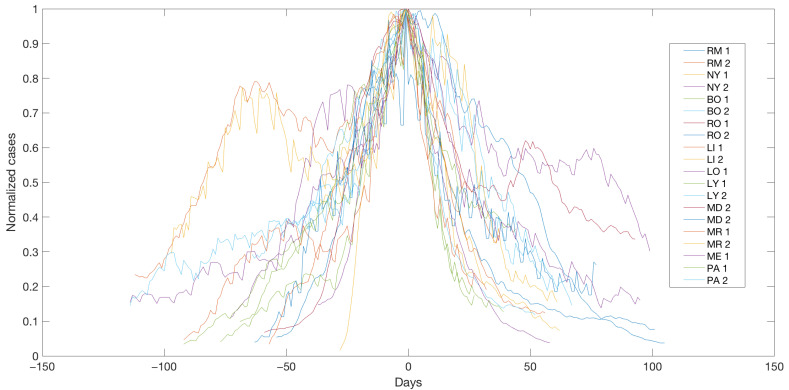
Normalization of reported case data.

**Figure 3 ijerph-19-16953-f003:**
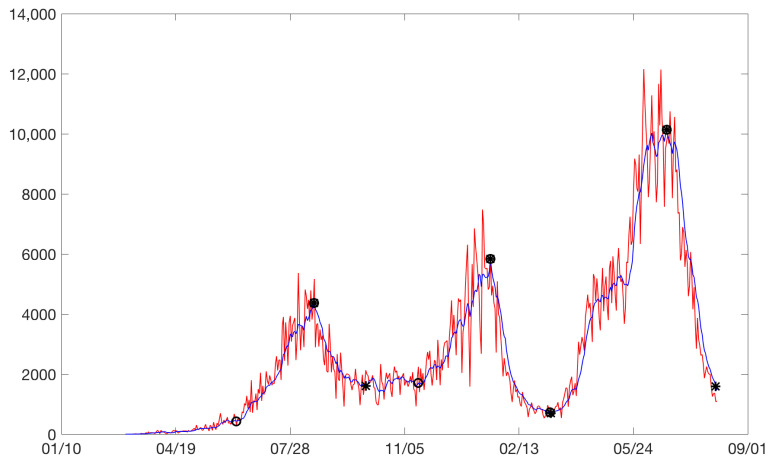
Bogota: on the horizontal axis is the date in days and on the vertical axis are the daily cases. Increases are identified with circles and drops with asterisks. Daily cases without moving average (red). Daily cases with 7 days moving average (blue).

**Figure 4 ijerph-19-16953-f004:**
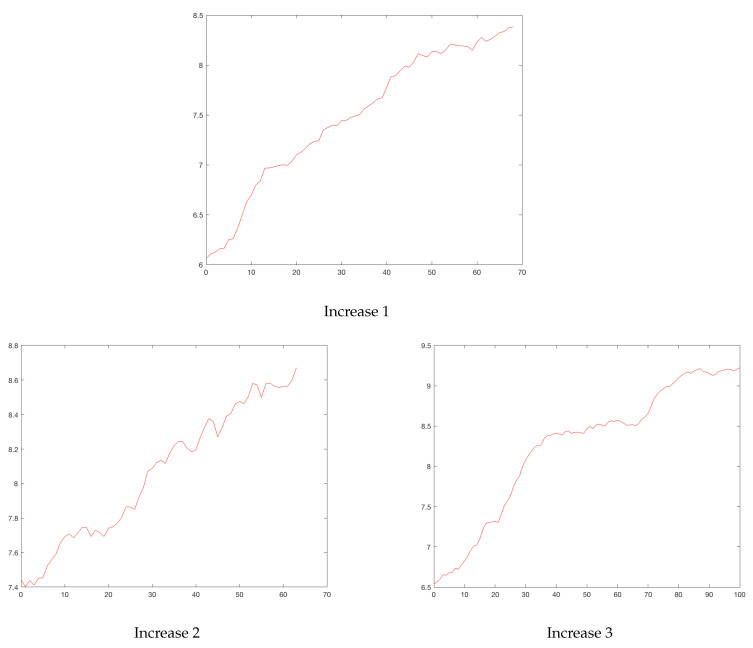
Bogota’s rises. Horizontal axis, days; vertical axis, logarithm of daily cases.

**Figure 5 ijerph-19-16953-f005:**
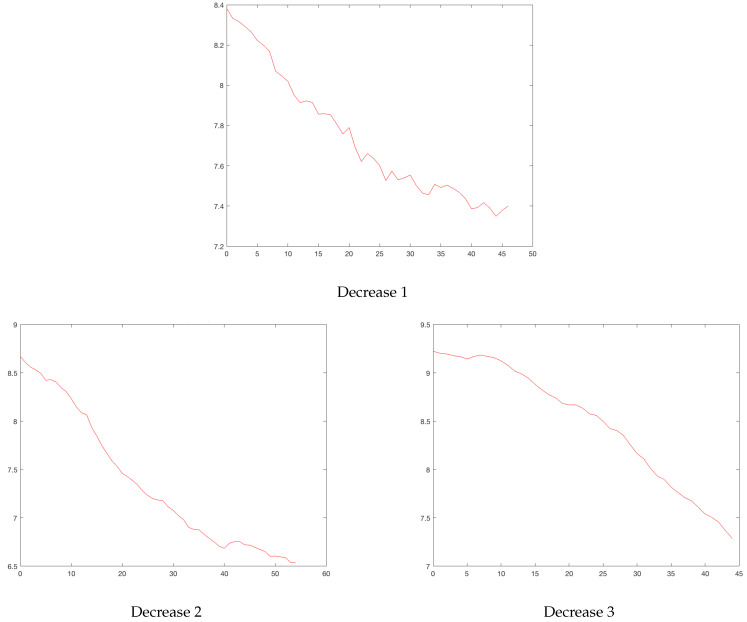
Bogota’s drops. Horizontal axis, days; vertical axis, logarithm of daily cases.

**Figure 6 ijerph-19-16953-f006:**
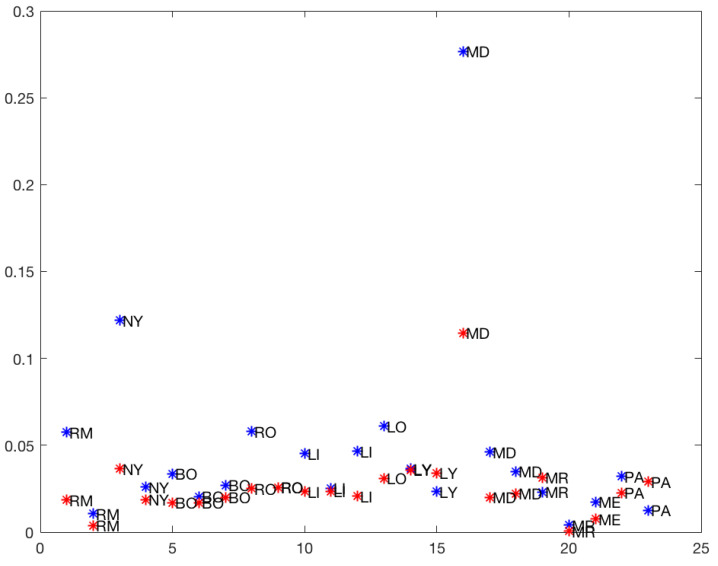
Upward slopes: blue: *total*, red: *50%*.

**Figure 7 ijerph-19-16953-f007:**
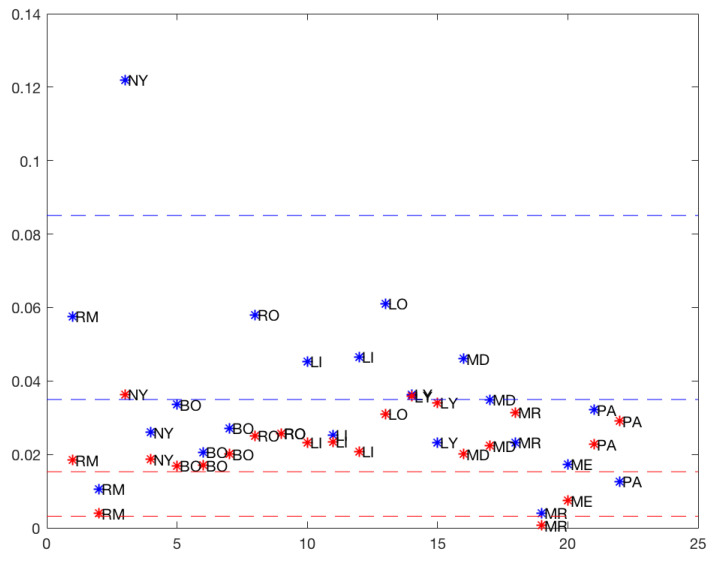
Cluster analysis for rises: blue: *total*, red: *50%*, (without MD).

**Figure 8 ijerph-19-16953-f008:**
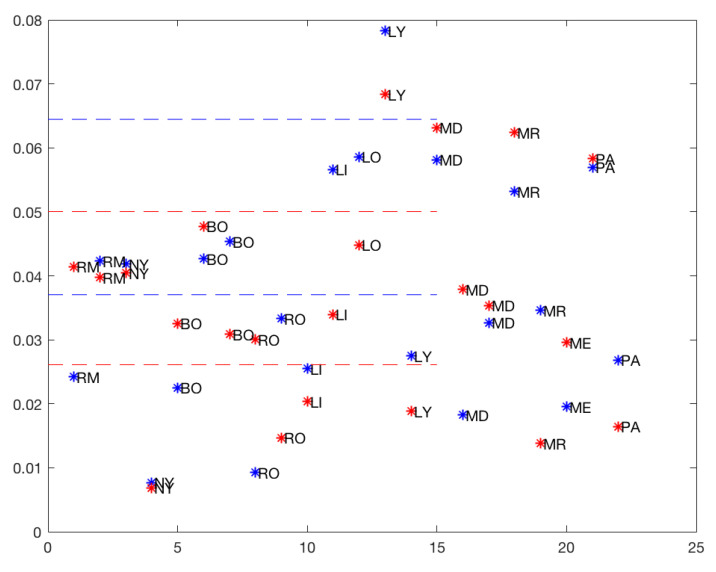
Downward slope: blue: *total*, red: *50%*.

**Figure 9 ijerph-19-16953-f009:**
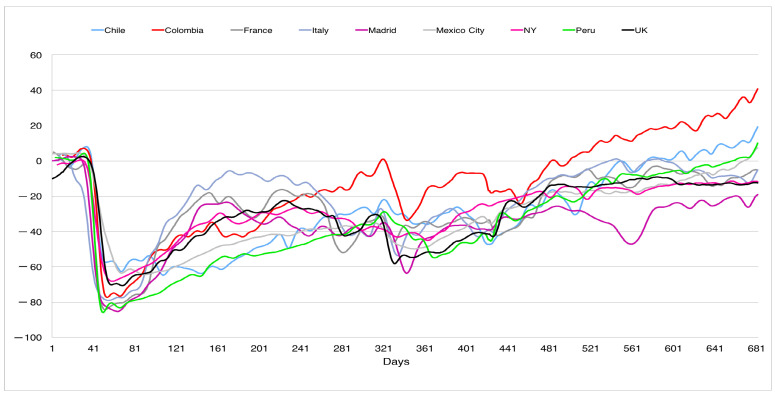
Social distancing in different countries’ cities between 4 April 2020 and 16 December 2021.

**Figure 10 ijerph-19-16953-f010:**
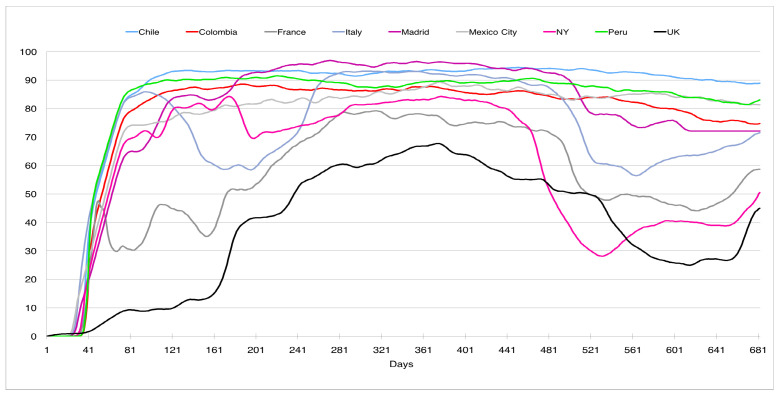
Use of mask in different countries’ cities between 4 April 2020 and 17 December 2021.

**Table 1 ijerph-19-16953-t001:** List of chosen cities.

City-Abbrevation	Access Date	Repositories
Bogota-BO	9 August 2021	http://saludata.saludcapital.gov.co/osb/index.php/datos-de-salud/enfermedades-trasmisibles/covid19/
Lima-LI	25 April 2021	https://covid19.minsa.gob.pe/sala_situacional.asp
London-LO	25 April 2021	https://coronavirus.data.gov.uk/details/deaths?areaType=region&areaName=London
Lyon-LY	10 June 2021	https://www.data.gouv.fr/fr/datasets/donnees-relatives-aux-resultats-des-tests-virologiques-covid-19/
Madrid-MD	14 June 2021	https://www.epdata.es/datos/evolucion-coronavirus-cada-comunidad/518/madrid/304
Marseille-MR	10 June 2021	https://www.data.gouv.fr/fr/datasets/donnees-relatives-aux-resultats-des-tests-virologiques-covid-19/
Mexico City-ME	25 April 2021	https://datos.covid-19.conacyt.mx/#DownZCSV
New York-NY	19 June 2021	https://www1.nyc.gov/site/doh/covid/covid-19-data.page
Paris-PA	10 June 2021	https://www.data.gouv.fr/fr/datasets/donnees-relatives-aux-resultats-des-tests-virologiques-covid-19/
Chilean Metropolitan Region-RM	8 August 2021	https://www.gob.cl/coronavirus/cifrasoficiales/#datos
Rome-RO	25 April 2021	https://github.com/pcm-dpc/COVID-19/blob/master/dati-regioni/dpc-covid19-ita-regioni.csv

**Table 2 ijerph-19-16953-t002:** Numbers of rises and drops for the eleven cities.

City	No. Increases	No. Decreases
BO	3	3
LI	3	2
LO	1	1
LY	2	2
MD	3	2
MR	2	2
ME	1	1
NY	2	2
PA	2	2
RM	2	2
RO	2	2

**Table 3 ijerph-19-16953-t003:** Estimation in the *total* regression model.

City	Peak	c = R, D	$|T_{\mathrm{ij}}-T_{\mathrm{ij}}^{c}|$	$\alpha_{\mathrm{ij}}^{c}$	*p*-Value	$\beta_{\mathrm{ij}}^{c}$	*p*-Value	$R^{2}$
BO	1	R	69	−2.05	$6.2\times 10^{59}$	0.0336	$1.1\times 10^{-46}$	0.95
BO	2	R	64	−1.26	$5.5\times 10^{-67}$	0.0206	$3.6\times 10^{-53}$	0.98
BO	3	R	103	−2.39	$2.7\times 10^{-71}$	0.0271	$8.8\times 10^{-55}$	0.91
LI	1	R	57	−2.28	$1.5\times 10^{-36}$	0.045	$5.9\times 10^{-27}$	0.88
LI	2	R	36	−0.82	$1.3\times 10^{-34}$	0.0253	$6.4\times 10^{-28}$	0.97
LI	3	R	47	−1.91	$5.5\times 10^{-36}$	0.047	$5.9\times 10^{-28}$	0.93
LO	1	R	20	−1.95	$7.1\times 10^{-30}$	0.061	$1.1\times 10^{-23}$	0.95
LY	1	R	77	−2.81	$1.2\times 10^{-66}$	0.036	$7.1\times 10^{-49}$	0.95
LY	2	R	46	−1.13	$3.4\times 10^{-31}$	0.023	$2.2\times 10^{-20}$	0.86
MD	1	R	11	−1.81	$1.5\times 10^{-04}$	0.27	0.001	0.81
MD	2	R	63	−2.56	$6.9\times 10^{-38}$	0.046	$2.0\times 10^{-27}$	0.86
MD	3	R	45	−1.58	$6.0\times 10^{-29}$	0.035	$2.7\times 10^{-19}$	0.85
MR	1	R	92	−2.25	$4.3\times 10^{-52}$	0.023	$1.9\times 10^{-31}$	0.78
MR	2	R	97	−0.73	$1.0\times 10^{-31}$	0.004	$5.4\times 10^{-07}$	0.23
ME	1	R	114	−2.01	$2.7\times 10^{-89}$	0.017	$9.5\times 10^{-63}$	0.92
NY	1	R	28	−2.55	$1.5\times 10^{-11}$	0.12	$5.5\times 10^{-09}$	0.74
NY	2	R	73	−1.84	$4.8\times 10^{-62}$	0.026	$2.4\times 10^{-46}$	0.95
PA	1	R	23	−2.77	$6.0\times 10^{-77}$	0.032	$2.5\times 10^{-58}$	0.95
PA	2	R	92	−1.69	$6.0\times 10^{-95}$	0.126	$1.0\times 10^{-61}$	0.92
RM	1	R	54	−2.62	$6.7\times 10^{-39}$	0.058	$1.6\times 10^{-30}$	0.92
RM	2	R	112	−1.13	$9.0\times 10^{-50}$	0.011	$1.5\times 10^{-30}$	0.7
RO	1	R	59	−2.92	$1.4\times 10^{-48}$	0.058	$7.2\times 10^{-39}$	0.95
RO	2	R	28	−0.65	$5.1\times 10^{-26}$	0.0257	$8.0\times 10^{-21}$	0.97
BO	1	D	46	−0.12	$1.5\times 10^{-06}$	−0.0225	$2.2\times 10^{-28}$	0.94
BO	2	D	54	−0.16	$8.2\times 10^{-04}$	−0.043	$4.8\times 10^{-34}$	0.94
BO	3	D	44	−0.26	$4.2\times 10^{-09}$	−0.045	$4.9\times 10^{-31}$	0.96
LI	1	D	39	−0.09	$1.5\times 10^{-02}$	−0.025	$2.1\times 10^{-18}$	0.88
LI	2	D	29	−0.13	$1.0\times 10^{-02}$	−0.057	$1.1\times 10^{-17}$	0.93
LO	1	D	60	0.13	$1.3\times 10^{-10}$	−0.586	$3.7\times 10^{-79}$	1
LY	1	D	29	−0.17	$4.2\times 10^{-04}$	−0.078	$4.3\times 10^{-22}$	0.97
LY	2	D	61	0.08	$2.0\times 10^{-03}$	−0.0275	$3.67\times 10^{-42}$	0.96
MD	1	D	48	−0.03	$1.8\times 10^{-01}$	−0.0581	$2.0\times 10^{-52}$	0.99
MD	2	D	79	−0.32	$7.0\times 10^{-16}$	−0.0183	$6.0\times 10^{-40}$	0.90
MD	3	D	55	−0.07	$3.1\times 10^{-02}$	−0.033	$1.7\times 10^{-36}$	0.95
MR	1	D	34	−0.05	$1.9\times 10^{-01}$	−0.053	$1.3\times 10^{-23}$	0.96
MR	2	D	63	−0.26	$5.1\times 10^{-07}$	−0.035	$9.3\times 10^{-36}$	0.92
ME	1	D	97	−0.11	$1.5\times 10^{-05}$	−0.0195	$4.5\times 10^{-66}$	0.96
NY	1	D	64	0.00	$5.5\times 10^{-107}$	−0.0419	$1.1\times 10^{-60}$	0.99
NY	2	D	101	−0.13	$5.6\times 10^{-09}$	−0.0077	$3.8\times 10^{-38}$	0.81
PA	1	D	41	−0.29	$5.7\times 10^{-01}$	−0.057	$4.2\times 10^{-26}$	0.95
PA	2	D	69	0.20	$4.3\times 10^{-10}$	−0.0268	$1.0\times 10^{-47}$	0.96
RM	1	D	103	−0.35	$1.4\times 10^{-11}$	−0.0242	$8.4\times 10^{-54}$	0.91
RM	2	D	58	0.04	0.1	−0.0423	$2.5\times 10^{-49}$	0.98
RO	1	D	95	−0.22	$2.5\times 10^{-11}$	−0.0093	$1.4\times 10^{-30}$	0.76
RO	2	D	107	−0.54	$2.1\times 10^{-18}$	−0.0333	$1.2\times 10^{-65}$	0.94

**Table 4 ijerph-19-16953-t004:** Estimation in the *50%* regression model.

City	Peak	c = r, d	$|T_{\mathrm{ij}}-T_{\mathrm{ij}}^{c}|$	$\alpha_{\mathrm{ij}}^{c}$	*p*-Value	$\beta_{\mathrm{ij}}^{c}$	*p*-Value	$R^{2}$
BO	1	r	29	−0.47	$7.2\times 10^{-22}$	0.017	$8.3\times 10^{-16}$	0.91
BO	2	r	35	−0.58	$2.4\times 10^{-30}$	0.017	$1.6\times 10^{-22}$	0.95
BO	3	r	44	−0.72	$1.4\times 10^{-25}$	0.02	$1.6\times 10^{-19}$	0.86
LI	1	r	28	−0.70	$4.6\times 10^{-22}$	0.023	$3.7\times 10^{-15}$	0.91
LI	2	r	29	−0.61	$2.4\times 10^{-25}$	0.023	$5.8\times 10^{-20}$	0.96
LI	3	r	25	−0.53	$1.2\times 10^{-18}$	0.021	$5.2\times 10^{-13}$	0.9
LO	1	r	20	−0.58	$1.7\times 10^{-15}$	0.031	$1.4\times 10^{-11}$	0.93
LY	1	r	19	−0.60	$1.0\times 10^{-10}$	0.036	$1.1\times 10^{-07}$	0.82
LY	2	r	19	−0.55	$7.3\times 10^{-14}$	0.034	$7.0\times 10^{-11}$	0.92
MD	1	r	6	−0.50	$1.9\times 10^{-03}$	0.11	0.007	0.87
MD	2	r	36	−0.85	$1.5\times 10^{-20}$	0.02	$2.1\times 10^{-11}$	0.74
MD	3	r	19	−0.49	$1.2\times 10^{-07}$	0.022	$6.6\times 10^{-04}$	0.50
MR	1	r	15	−0.41	$1.8\times 10^{-06}$	0.031	$1.7\times 10^{-04}$	0.67
MR	2	r	82	−0.50	$1.1\times 10^{-21}$	0.0006	0.45	0.07
ME	1	r	43	−0.50	$2.8\times 10^{-15}$	0.007	$5.9\times 10^{-05}$	0.33
NY	1	r	18	−0.52	$2.8\times 10^{-11}$	0.036	$6.5\times 10^{-09}$	0.89
NY	2	r	36	−0.76	$1.6\times 10^{-27}$	0.019	$7.8\times 10^{-18}$	0.89
PA	1	r	23	−0.55	$3.8\times 10^{-16}$	0.023	$9.9\times 10^{-11}$	0.87
PA	2	r	36	−0.81	$1.0\times 10^{-10}$	0.029	$9.3\times 10^{-17}$	0.93
RM	1	r	25	−0.48	$4.4\times 10^{-15}$	0.018	$1.3\times 10^{-09}$	0.80
RM	2	r	84	−0.49	$2.9\times 10^{-30}$	0.0039	$1.1\times 10^{-09}$	0.37
RO	1	r	25	−0.51	$3.5\times 10^{-15}$	0.025	$8.2\times 10^{-12}$	0.87
RO	2	r	28	−0.64	$5.1\times 10^{-26}$	0.026	$8.0\times 10^{-21}$	0.97
BO	1	d	23	−0.01	$3.4\times 10^{-01}$	−0.033	$4.4\times 10^{-19}$	0.98
BO	2	d	15	0.01	$4.7\times 10^{-01}$	−0.048	$2.5\times 10^{-11}$	0.97
BO	3	d	26	0.10	$2.8\times 10^{-04}$	−0.031	$1.0\times 10^{-15}$	0.94
LI	1	d	24	−0.14	$1.4\times 10^{-02}$	−0.02	$2.9\times 10^{-05}$	0.56
LI	2	d	18	−0.02	$5.6\times 10^{-01}$	−0.034	$1.0\times 10^{-08}$	0.88
LO	1	d	17	0.02	$1.2\times 10^{-01}$	−0.045	$1.7\times 10^{-14}$	0.98
LY	1	d	13	0.14	$5.5\times 10^{-02}$	−0.068	$1.6\times 10^{-05}$	0.83
LY	2	d	31	−0.02	$4.7\times 10^{-01}$	−0.019	$6.1\times 10^{-13}$	0.84
MD	1	d	13	−0.00	$8.8\times 10^{-01}$	−0.063	$2.4\times 10^{-09}$	0.97
MD	2	d	17	−0.12	$2.1\times 10^{-02}$	−0.038	$9.3\times 10^{-07}$	0.81
MD	3	d	19	0.0	$9.9\times 10^{-01}$	−0.035	$1.2\times 10^{-08}$	0.86
MR	1	d	12	0.05	$2.9\times 10^{-01}$	−0.062	$1.5\times 10^{-06}$	0.91
MR	2	d	31	0.00	$9.8\times 10^{-01}$	−0.014	$1.5\times 10^{-06}$	0.56
ME	1	d	24	0.07	$2.8\times 10^{-02}$	−0.03	$8.8\times 10^{-12}$	0.88
NY	1	d	19	0.03	$9.2\times 10^{-02}$	−0.04	$3.0\times 10^{-15}$	0.98
NY	2	d	84	−0.16	$5.3\times 10^{-11}$	−0.0068	$6.4\times 10^{-26}$	0.74
PA	1	d	13	0.08	$9.0\times 10^{-02}$	−0.058	$8.9\times 10^{-07}$	0.90
PA	2	d	34	0.03	$1.7\times 10^{-01}$	−0.016	$3.8\times 10^{-14}$	0.8
RM	1	d	20	0.08	$2.0\times 10^{-04}$	−0.041	$9.1\times 10^{-16}$	0.97
RM	2	d	19	0.07	$4.5\times 10^{-04}$	−0.040	$1.0\times 10^{-14}$	0.97
RO	1	d	26	0.08	$9.9\times 10^{-07}$	−0.03	$1.0\times 10^{-22}$	0.98
RO	2	d	51	0.07	$1.6\times 10^{-06}$	−0.0147	$3.8\times 10^{-35}$	0.96

**Table 5 ijerph-19-16953-t005:** Upward slopes for *total* and *50%* models.

		*Total*		*50*%		
City	Peak	Days	Slope	Days	Slope	Days × Slope
BO	1	68	0.034	29	0.017	0.49
BO	2	63	0.021	35	0.017	0.60
BO	3	100	0.027	44	0.02	0.88
LI	1	56	0.045	28	0.023	0.64
LI	2	35	0.025	29	0.023	0.67
LI	3	46	0.047	25	0.021	0.53
LO	1	36	0.061	20	0.031	0.62
LY	1	76	0.036	19	0.036	0.68
LY	2	45	0.023	19	0.034	0.65
MD	1	8	0.277	6	0.115	0.69
MD	2	62	0.046	36	0.02	0.72
MD	3	44	0.035	19	0.022	0.42
MR	1	91	0.023	15	0.031	0.47
MR	2	96	0.004	82	0.001	0.08
ME	1	113	0.017	43	0.007	0.30
NY	1	27	0.122	18	0.036	0.65
NY	2	72	0.026	36	0.019	0.68
PA	1	91	0.032	23	0.023	0.53
PA	2	113	0.014	29	0.029	0.84
RM	1	53	0.058	25	0.018	0.45
RM	2	111	0.011	84	0.004	0.34
RO	1	58	0.058	25	0.025	0.63
RO	2	27	0.026	28	0.026	0.73

**Table 6 ijerph-19-16953-t006:** Cluster analysis for rises at total and *50%*, without the first rise of MD.

		*50*%			*Total*		
**City**	**Peak**	**Class**	**Mean**	**Slope**	**Class**	**Mean**	**Slope**
RM	2	1	0.004	0.004	1	0.022	0.011
MR	2	1	0.004	0.001	1	0.022	0.004
ME	1	1	0.004	0.007	1	0.022	0.017
RM	1	2	0.021	0.018	2	0.048	0.058
NY	2	2	0.021	0.019	1	0.022	0.026
BO	1	2	0.021	0.017	1	0.022	0.034
BO	2	2	0.021	0.017	1	0.022	0.021
BO	3	2	0.021	0.02	1	0.022	0.027
RO	1	2	0.021	0.025	2	0.048	0.058
RO	2	2	0.021	0.026	1	0.022	0.026
LI	1	2	0.021	0.023	2	0.048	0.045
LI	2	2	0.021	0.023	1	0.022	0.025
LI	3	2	0.021	0.021	2	0.048	0.047
MD	2	2	0.021	0.02	2	0.048	0.046
MD	3	2	0.021	0.022	1	0.022	0.035
PA	1	2	0.021	0.023	1	0.022	0.032
NY	1	3	0.033	0.036	3	0.122	0.122
LO	1	3	0.033	0.031	2	0.048	0.061
LY	1	3	0.033	0.036	1	0.022	0.036
LY	2	3	0.033	0.034	1	0.022	0.023
MR	1	3	0.033	0.031	1	0.022	0.023
PA	2	3	0.033	0.029	1	0.022	0.013

**Table 7 ijerph-19-16953-t007:** Downward slopes for *total* and *50%* models.

		*Total*		*50*%		
City	Peak	Days	Slope	Days	Slope	−Days × Slope
BO	1	46	−0.023	23	−0.033	0.76
BO	2	54	−0.043	15	−0.048	0.72
BO	3	44	−0.045	26	−0.031	0.81
LI	1	38	−0.025	24	−0.02	0.48
LI	2	28	−0.057	18	−0.034	0.61
LO	1	59	−0.059	17	−0.045	0.77
LY	1	28	−0.078	13	−0.068	0.88
LY	2	60	−0.027	31	−0.019	0.59
MD	1	47	−0.058	13	−0.063	0.82
MD	2	78	−0.018	17	−0.038	0.65
MD	3	54	−0.033	19	−0.035	0.67
MR	1	33	−0.053	12	−0.062	0.74
MR	2	62	−0.035	31	−0.014	0.43
ME	1	96	−0.02	24	−0.03	0.72
NY	1	63	−0.042	19	−0.04	0.76
NY	2	100	−0.008	84	−0.007	0.59
PA	1	40	−0.057	13	−0.058	0.75
PA	2	68	−0.027	34	−0.016	0.54
RM	1	102	−0.024	20	−0.041	0.82
RM	2	57	−0.042	19	−0.04	0.76
RO	1	94	−0.009	26	−0.03	0.78
RO	2	106	−0.033	51	−0.015	0.77

**Table 8 ijerph-19-16953-t008:** Cluster analysis for drops at total and 50%.

		*50*%			*Total*		
**City**	**Peak**	**Class**	**Mean**	**Slope**	**Class**	**Mean**	**Slope**
LY	1	1	−0.063	−0.068	1	−0.078	−0.078
MD	1	1	−0.063	−0.063	2	−0.051	−0.058
MR	1	1	−0.063	−0.062	2	−0.051	−0.053
PA	1	1	−0.063	−0.058	2	−0.051	−0.057
RM	1	2	−0.037	−0.041	3	−0.023	−0.024
RM	2	2	−0.037	−0.04	2	−0.051	−0.042
NY	1	2	−0.037	−0.04	2	−0.051	−0.042
BO	1	2	−0.037	−0.033	3	−0.023	−0.023
BO	2	2	−0.037	−0.048	2	−0.051	−0.043
BO	3	2	−0.037	−0.031	2	−0.051	−0.045
RO	1	2	−0.037	−0.03	3	−0.023	−0.009
LI	2	2	−0.037	−0.034	2	−0.051	−0.057
LO	1	2	−0.037	−0.045	2	−0.051	−0.059
MD	2	2	−0.037	−0.038	3	−0.023	−0.018
MD	3	2	−0.037	−0.035	3	−0.023	−0.033
ME	1	2	−0.037	−0.03	3	−0.023	−0.02
NY	2	3	−0.015	−0.007	3	−0.023	−0.008
RO	2	3	−0.015	−0.015	3	−0.023	−0.033
LI	1	3	−0.015	−0.02	3	−0.023	−0.025
LY	2	3	−0.015	−0.019	3	−0.023	−0.027
MR	2	3	−0.015	−0.014	3	−0.023	−0.035
PA	2	3	−0.015	−0.016	3	−0.023	−0.027

## Data Availability

Data and models that support the findings of this study are available from the corresponding author.

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
