# Peer review of "COVID-19: A Comparative Study of Contagions Peaks in Cities from Europe and the Americas"

_ijerph, 2022, doi:10.3390/ijerph192416953_

Round 1
Reviewer 1 Report (Previous Reviewer 1)
Despite being a fairly simple model, and the information from the cities analyzed is not homogeneous, for example, it only analyzes a peak in London and Mexico City, the information is interesting in the sense that it finds similarities between different cities with different geographic locations and economic development.
This manuscript is a resubmission of an earlier submission. The following is a list of the peer review reports and author responses from that submission.
Round 1
Reviewer 1 Report
I apologize in advance for my biomedical training, and for which the way of structuring an article is based on a format that I feel does not comply with this document.
In the abstract:
Missing information, it must contain.
Background: the context and purpose of the study
Methods: how the study was performed and statistical tests used
Results: the main findings
Conclusions: brief summary and potential implications.
The introduction should be limited to:
What is known about the topic:
What are the knowledge gaps that have been identified and the implications of addressing the knowledge gaps.
Information necessary to understand why the objective is established.
And finally the objective of the study, or sets the goal at the beginning of the methodology.
The following paragraph corresponds to methodology:
To compare the development of different cities, with different control measures, we have found that normalizing them to the maximum observed at each peak makes the development comparable. This implies a certain self-similarity of the processes involved. The differences in behavior observed after normalization may be due to the effects of the various controls, i.e., the processes are not entirely self-similar.
The last paragraph of the introduction should be located in methodology:
Therefore, we use a regression model where the response variable is the logarithm of daily infected cases. Then we estimate the uphill and downhill slopes, with the whole data and with only the data to reach the 50% of the rise/down of each peak. We perform a cluster analysis using estimated uphill and downhill slopes that put in evidence different behaviors between cities according to the peaks. In Section 2.1, we present the information about chosen cities. In Section 2.2 the regression model for the log-data is explained. The study of the rising slopes and the cluster analysis are explained in Section 3.1, similar results for the down slopes are given in Section 3.2. Conclusions about our analysis are given in Section 4.
In methodology.
It does not establish the temporality of the study although reading the article it is understood.
On page 3 they set the goal:
In this paper we want to compare the cities with each other, in terms of the downward/upward slope. Since the 11 cities have different size populations, the number of new daily cases are not on the same scale. Moreover these data are highly non-linear. Therefore, we normalize the data by applying the natural logarithm function.
On page 4 includes interpretation or utility of the data that should be addressed in the discussion:
Our study also serves to understand whether cities in the southern hemisphere that had their Covid peaks after those in Europe anticipated the measures that should have been taken. Of course the height of the peaks strongly depends on several reasons, namely climatic conditions, confinement measures, testing intensity among many others. However, the evidence provided by the data shows that when the curves of detected cases are rescaled they have an unexpected regularity and similarity between different cities of the world with disparate socioeconomic conditions, with very different health systems and where very different containment measures were applied. In this direction, Figure 3 shows the evolution of the social distancing, measured through cell phone data, in different geographical areas, showing a higher variability (except for the first peak) than our normalized curves. It is interesting to note how this distancing has evolved upwards after the first wave, with smaller declines in the periods of the successive peaks. This disparity is also observed in the use of masks (see Figure 4) in public places for the different geographical areas. To this must be added the high unpredictability of the evolution of Covid, as shown in Canals et al. [9], where a comparison of the main Lyapunov exponent is made on the same data series for different countries, obtaining that these vary between 5 and 15, being much higher than in other epidemics such as N1H1, where the main coefficient is of the order of 2.
This is how the discussion should start (in the document it is in conclusions):
In this article, we have provided a simple statistical analysis of the new Coronavirus (COVID-19) outbreak in 11 cities from Europe and America. Using daily infected data in the cities for approximately the months after the first cases were confirmed in each city, we have analysed downward and rises slopes per each wave using the log-linear model approach. These models are consistent with the ups and downs of SIR-type models if one thinks about each of the pandemic waves.
In discussion:
discuss why?
Our study also serves to understand whether the cities in the southern hemisphere that had their covid peaks after those in Europe anticipated the measures that should have been taken. In addition, we found that rises and downs are almost independent of the type of quarantine measures taken, more or less restrictive.
This section should discuss the implications of the findings in context of existing research and highlight limitations of the study.
Reviewer 2 Report
The abstract is too short, add some of the conclusion to the abstract
Figure 1 needs colour amendments: example, (ME,BO) look similar, please change. Also, the colours, (PA,RQ), (MD,RM) have similar issue
Figures Rise 1, Rise 2, Rise 3, down 1, down 2….etc the lines and the axis’s look so faint, make it clearer
Tables caption should be at the top of the table instead of the table bottom
Page number 3: the chapter of “In this paper we want to compare the cities” needs revising in choosing words, and punctuations.
Use the passive voice or simple present but avoid using the “we”, example: we noramlise the data applying….
Change it to the data is normalized.
Another example: we denote N(t), to N(t) denotes the daily number of infected……
Figure 2: increase the font size for the captions “days” on x-axis, and “normalized cases” on y-axis
This sentence is not clear, please re-phrase “Our study also serves to understand whether cities in the southern hemisphere that had their Covid
peaks after those in Europe anticipated the measures that should have been taken”
For figure 2 and normalization, please explain in a small paragraph the meaning of the centred normalized raw data and give an example of real life. Readers need to understand the real purport behind the statistical phrases.
Please replace mask with “face mask”
Figure 3 does not show any axis caption. What is the meaning of minus sign in y-axis
Please explain Figure 4 and add axis caption.
The whole section needs revising because it is hard to follow with your various diagrams with the meaning and the link between them.
In page 6, I know that you already explained the meaning of asterisk (the end of the downs), but it is a good idea to explain this in page 6 as well, together with equations 1, and 2, such as “We use the notation R = rise, D = down, and *=end of down for the total mode”
Add one or two sentences explaining how you derived alpha, and beta in equations (3) and (4)
In Page 10, “In this section we study the slopes, both total and at 50%, and we will classify”. You need to use “will” in both sentences, or do not use it, and much better to keep it as a passive voice. So the sentence can be “In this section, the slop was studies in both regression models, total and 50%. Cities are classified according to ……”
The conclusion is too long, remove one or two paragraphs
Reviewer 3 Report
The following Chapters must be improved:
Methods, Discussions and Conclusions
with more information about the subject
and also giving more data on the subject from the vadt literature.
English mush be improved also.
Author Response
Thank you to the referee that gives us several observations that allows us to improve the manuscript.
- The following Chapters must be improved: Methods, Discussions and Conclusions with more information about the subject and also giving more data on the subject from the vast literature.
We revise deeply the indicated sections.
- English mush be improved also.
We revise the English.